# Diagnostic utility of the amyotrophic lateral sclerosis Functional Rating Scale—Revised to detect pharyngeal dysphagia in individuals with amyotrophic lateral sclerosis

Jennifer L. Chapin[1], Lauren Tabor Gray[1,2], Terrie Vasilopoulos[3], Amber Anderson[1,4], Lauren DiBiase[1,4], Justine Dallal York[1,4], Raele Robison[1,4], James Wymer[5], Emily K. Plowman[1,4,5,6]*

1 Aerodigestive Research Core, University of Florida, Gainesville, FL, United States of America, 2 Department of Neurology, Phil Smith Neuroscience Institute, Holy Cross Hospital, Fort Lauderdale, FL, United States of America, 3 Department of Anesthesiology, College of Medicine, University of Florida, Gainesville, FL, United States of America, 4 Department of Speech, Language and Hearing Sciences, University of Florida, Gainesville, FL, United States of America, 5 Department of Neurology, College of Medicine, University of Florida, Gainesville, FL, United States of America, 6 Department of Surgery, College of Medicine, University of Florida, Gainesville, FL, United States of America

* eplowman@phhp.ufl.edu

## Abstract

### Objective

The ALS Functional Rating Scale–Revised (ALSFRS-R) is the most commonly utilized instrument to index bulbar function in both clinical and research settings. We therefore aimed to evaluate the diagnostic utility of the ALSFRS-R bulbar subscale and swallowing item to detect radiographically confirmed impairments in swallowing safety (penetration or aspiration) and global pharyngeal swallowing function in individuals with ALS.

### Methods

Two-hundred and one individuals with ALS completed the ALSFRS-R and the gold standard videofluoroscopic swallowing exam (VFSE). Validated outcomes including the Penetration-Aspiration Scale (PAS) and Dynamic Imaging Grade of Swallowing Toxicity (DIGEST) were assessed in duplicate by independent and blinded raters. Receiver operator characteristic curve analyses were performed to assess accuracy of the ALSFRS-R bulbar subscale and swallowing item to detect radiographically confirmed unsafe swallowing (PAS > 3) and global pharyngeal dysphagia (DIGEST >1).

### Results

Although below acceptable screening tool criterion, a score of $\leq 3$ on the ALSFRS-R swallowing item optimized classification accuracy to detect global pharyngeal dysphagia (sensitivity: 68%, specificity: 64%, AUC: 0.68) and penetration/aspiration (sensitivity: 79%, specificity: 60%, AUC: 0.72). Depending on score selection, sensitivity and specificity of the

**Data Availability Statement:** All relevant data are within the paper and its supporting information files.

**Funding:** E.P. received research grant 1R01NS100859-01 from the National Institute of Neurological Disorders and Stroke (NINDS). https://projectreporter.nih.gov/project_info_description.cfm?projectnumber=1R01NS100859-01 E.P. also received a clinical management grant, 17-CM-323 from the ALS Association. http://www.alsa.org/research/ The sponsors did not play any role in the study design, data collection and analysis, decision to publish, or preparation of the manuscript.

**Competing interests:** The authors have declared that no competing interests exist.

ALSFRS-R bulbar subscale ranged between 34–94%. A score of < 9 optimized classification accuracy to detect global pharyngeal dysphagia (sensitivity: 68%, specificity: 68%, AUC: 0.76) and unsafe swallowing (sensitivity:78%, specificity:62%, AUC: 0.73).

## Conclusions

The ALSFRS-R bulbar subscale or swallowing item did not demonstrate adequate diagnostic accuracy to detect radiographically confirmed swallowing impairment. These results suggest the need for alternate screens for dysphagia in ALS.

## Introduction

Amyotrophic lateral sclerosis (ALS) is a progressive and fatal neurodegenerative disease affecting both upper and lower motor neurons within the cortex, brainstem and spinal cord [1]. Dysphagia, or swallowing impairment, occurs in a reported 85% of patients with ALS at some point during the disease process and is associated with malnutrition, weight loss, reduced quality of life, aspiration pneumonia and death [2–6]. Early detection and consistent monitoring of dysphagia provides the opportunity to mitigate associated risks and improve survival with timely interventions [7].

A universally accepted and validated clinical test battery to accurately assess and monitor bulbar disease progression is currently lacking [8]. A 2017 survey of Northeast ALS (NEALS) centers in the United States revealed highly variable practice patterns for the evaluation of bulbar function in patients with ALS [9]. Both clinical and instrumental swallow evaluations were found to be underutilized in multidisciplinary ALS clinics with less than 60% of respondents utilizing clinical swallow assessments and only 27% referring for the gold standard videofluoroscopic swallowing evaluation. Importantly, this survey revealed that the only clinical test *routinely* performed to evaluate bulbar function (>90% of sites) was the revised ALS Functional Rating Scale (ALSFRS-R).

The ALSFRS-R is a 12-item questionnaire with each question rated on a 5-point ordinal scale used to monitor progression of disability in patients with ALS. The scale currently represents the most widely used ALS outcome measure in Phase II and III clinical trials and longitudinal studies [10]. More recently, the psychometric properties of the ALSFRS-R have been evaluated with evidence suggesting multidimensionality, and the utilization of individual subscale scores rather than a total score has been recommended [11–13]. These bulbar, motor and respiratory subscores are intended to provide more precise prognostic information, as the individual's domain scores have been demonstrated to be more clinically robust when reported as subscores rather than a combined score [14]. Specifically, individuals with bulbar-onset disease demonstrated slower rate of decline on the motor subscore and hastened decline on the bulbar subscore compared to those with spinal onset disease [14]. One study of 18 individuals with motor neuron disease (MND) investigated the relationship between the ALSFRS-R bulbar subscore and radiographically confirmed airway invasion; however, individuals who aspirated during swallowing were not included in the study cohort, limiting the generalization of the results [15]. The discriminant ability of the bulbar subscore and swallowing item score to detect radiographically confirmed pharyngeal dysphagia in ALS has not yet been determined. We therefore sought to evaluate the discriminant ability of the ALSFRS-R bulbar subscale and swallowing item scores to classify early radiographically confirmed pharyngeal dysphagia in patients with ALS. Given the that the scale is a five-point ordinal scale that lacks

linearity, we hypothesized that the ALSFRS-R would not demonstrate adequate sensitivity to detect mild changes in pharyngeal swallowing function in individuals with ALS.

## Materials and methods

### Participants

All eligible ALS patients who attended the University ALS clinic were informed of the study and invited to participate, representing a convenience sample. Two-hundred and one individuals were enrolled in this study. Inclusion criteria were: 1) confirmed diagnosis of ALS (Revised El Escorial criteria) by a neuromuscular neurology specialist; 2) not pregnant, 3) no allergies to barium, and 4) still consuming some form of foods and liquids by mouth. This study was approved by the University of Florida Institutional Review Board and conducted in accordance with the Declaration of Helsinki. All participants provided informed written consent. Participants attended a single testing session which included completion of the ALSFRS-R (index test) and a standardized videofluoroscopic swallowing examination (VFSE, gold standard reference test).

**Index test.**   The ALSFRS-R is a 12-item questionnaire validated to monitor functional disease progression across four subscales of activities of daily living that include bulbar, fine motor, gross motor and respiratory domains [10]. Three items assessing speech, salivation, and swallowing comprise the bulbar subscale with each item scored on a five-point ordinal scale (0–4) for a total of 12 points, with higher scores indicating better self-reported function (0 = total loss of function, 12 = normal functioning). A single question on swallowing is scored as follows 4—Normal eating habits, 3—Early eating problems; occasional choking, 2—Dietary consistency changes, 1—Needs supplemental tube feeding, 0—NPO (exclusively parenteral or enteral feeding) [10]. Participants completed the ALSFRS-R in interview fashion with an opportunity for input by their caregivers. All research personnel conducting these interviews completed training in the administration and scoring of the ALSFRS-R.

**Reference standard.**   VFSE was completed by a trained research speech-language pathologist (SLP) with participants comfortably seated in an upright lateral viewing plane using a properly collimated Phillips BV Endura fluoroscopic C-arm unit (GE 9900 OEC Elite Digital Mobile C-Arm system type 718074). Fluoroscopic images and synced audio were digitally recorded at 30 frames per second using a high resolution (1024 x 1024) TIMS DICOM system (Version 3.2, TIMS Medical, TM, Chelmsford, MA) for subsequent analysis. A standardized bolus presentation was administered utilizing a cued instruction to swallow and included: three 5 mL thin liquid barium, one comfortable cup sip of thin liquid barium, three 5 mL thin honey barium, two 5 mL pudding consistency barium, and a ¼ graham cracker square coated with pudding consistency barium (Varibar®, Bracco Diagnostics, Inc., Monroe Township, NJ). If the patient was unable to self-feed due to upper extremity weakness, clinician assistance or alternative methods (i.e., straw) were employed, consistent with the individual's feeding methods routinely utilized at home. SLPs enforced standardized bailout criteria requiring administration of thicker consistencies following two instances of aspiration and discontinuation if an additional aspiration event occurred during the exam. VFSE recordings were saved to a secure server and blinded for subsequent analysis.

### VFSE outcome measures

Each VFSE was rated in duplicate by two trained, blinded and independent raters. Complete agreement (100%) was required for all ratings, with a discrepancy meeting utilized to finalize any inconsistent ratings between raters.

| | PAS Score Description: | Safety Status: | |
|---|---|---|---|
| 1 | Material does not enter the airway | Safe | |
| 2 | Material enters the airway, remains above the vocal folds, is ejected from airway | | |
| 3 | Material enters the airway, remains above vocal folds, is not ejected from airway | Unsafe | |
| 4 | Material enters the airway, contacts the vocal folds, is ejected from airway | | |
| 5 | Material enters the airway, contacts the vocal folds, is not ejected from airway | | |
| 6 | Material enters the airway, passes below vocal folds, ejected from trachea | | |
| 7 | Material enters the airway, passes below vocal folds, not ejected despite effort | | |
| 8 | Material enters the airway, passes below the vocal folds, no effort made to eject | | |

**Fig 1. Penetration Aspiration Scale (PAS) scores and corresponding binary classifications of swallowing safety status.** A) Representative videofluoroscopic images depicting safe swallowing with no contrast material entering the airway. B) Laryngeal penetration. C) Tracheal aspiration [16].

**Swallowing safety.** The Penetration-Aspiration Scale (PAS) was utilized to evaluate swallowing safety (Fig 1) [16]. This validated eight-point ordinal scale indexes the depth of contrast material entering the airway during swallowing events, the presence of a protective response, and if aspirate material was ejected from the airway [16]. All elicited swallows within a given bolus trial were ascribed a PAS score and the worst PAS score utilized for statistical analysis. Fig 1 denotes the PAS with established categorical levels of airway safety used.

**Global pharyngeal swallowing.** The dynamic imaging grade of swallowing toxicity (DIGEST) is a validated five-point ordinal scale created to assess both efficiency and safety of bolus flow [17] and recently utilized in ALS [18]. The DIGEST (Fig 2) yields a global grade of

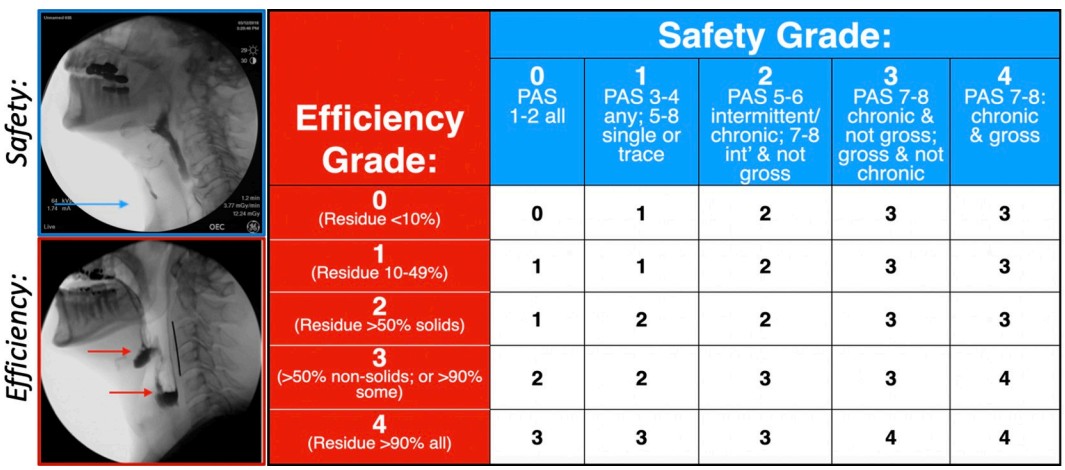

**Fig 2. Dynamic Imaging Grade of Swallowing Toxicity (DIGEST) outcome [17].**

pharyngeal dysphagia evaluated on bolus transport during the entirety of the videofluoro-scopic swallow study to determine clinically relevant categories of overall pharyngeal dyspha-gia severity levels. DIGEST total scores are a composite of two subscores (scored 0–4) addressing: (i) swallowing efficiency based on degree of bolus clearance, and (ii) airway safety based on severity and frequency of PAS scores. DIGEST scores of zero indicate normal swal-lowing while total and subscore grades of 4 indicate life-threatening dysphagia.

**Statistical analysis.** Descriptives were performed to summarize participant demographics and outcomes of interests. A receiver operator characteristic (ROC) curve analysis was then performed on the index test (ALSFRS-R bulbar subscale and swallowing item) to identify unsafe (PAS > 3) and global dysphagia (DIGEST > 1). Area under the curve (AUC) with boot-strapped 95% confidence intervals, sensitivity, specificity, positive predictive value (PPV), and negative predictive values (NPV) were calculated using JMP Pro Version 14.1.0 (SAS Institute Inc., Cary, NC). Optimal classification cutoffs for index test were determined by values that maximized both sensitivity and specificity.

## Results

### Participant demographics

Complete ALSFRS-R and VFSE data were missing in four participants resulting in 197 patients in the final analysis. Mean age was 62.9 ($SD = 10.3$) and average ALS disease duration was 26.6 months from symptom onset ($SD = 23.6$). Fifty-three percent were male ($n = 106$) and 58.1% presented with a spinal onset ($n = 111$). Mean ALSFRS-R score was 35.3 ($SD = 7.4$). Frequency data for the ALSFRS-R bulbar subscale and swallowing item scores are presented in histogram plots in Fig 3A and 3B respectively. Mean ALSFRS-R bulbar subscale score was 9.1 ($SD = 2.4$) and mean swallowing item score was 3.05 ($SD = 0.79$). Radiographically confirmed unsafe swallowing was identified in 38.9% of patients ($n = 76$, Fig 3C) and prevalence of global pha-ryngeal dysphagia was 58.9% ($n = 116$, Fig 3D).

### Discriminant ability of the ALSFRS-R to detect swallowing impairment

Scatterplots depicting relationships between the ALSFRS-R and swallowing outcomes of inter-est are shown in Fig 4. ROC curve results for the ALSFRS-R bulbar subscale and ALSFRS-R swallowing items to detect radiographically confirmed penetrators/aspirators are presented in Table 1 and Fig 5A and 5B. Classification accuracy for both ALSFRS-R outcomes to detect global pharyngeal dysphagia are presented in Table 2 and Fig 5C and 5D. Optimized classifica-tion cutoff of ALSFRS-R swallowing score of ≤ 3 and ALSFRS-R bulbar score of ≤ 9 were found for both outcomes.

## Discussion

To our knowledge, this represents the first investigation to compare ALSFRS-R bulbar out-comes to those of the gold standard reference test for swallowing (VFSE). The ALSFRS-R bul-bar subscale and swallowing item demonstrated poor to fair diagnostic accuracy to detect radiographically confirmed pharyngeal swallowing impairment in the 197 ALS patients exam-ined (AUC: 0.68–0.76). No cut score emerged for ALSFRS-R outcome with an acceptable level of classification accuracy to distinguish normal versus disordered swallowing. Thus, the ALSFRS-R did not demonstrate adequate clinical utility as a screening tool to detect early pha-ryngeal dysphagia and demonstrated insufficient sensitivity as a marker of change in pharyn-geal swallowing function for research clinical trials. These findings highlight the need for the

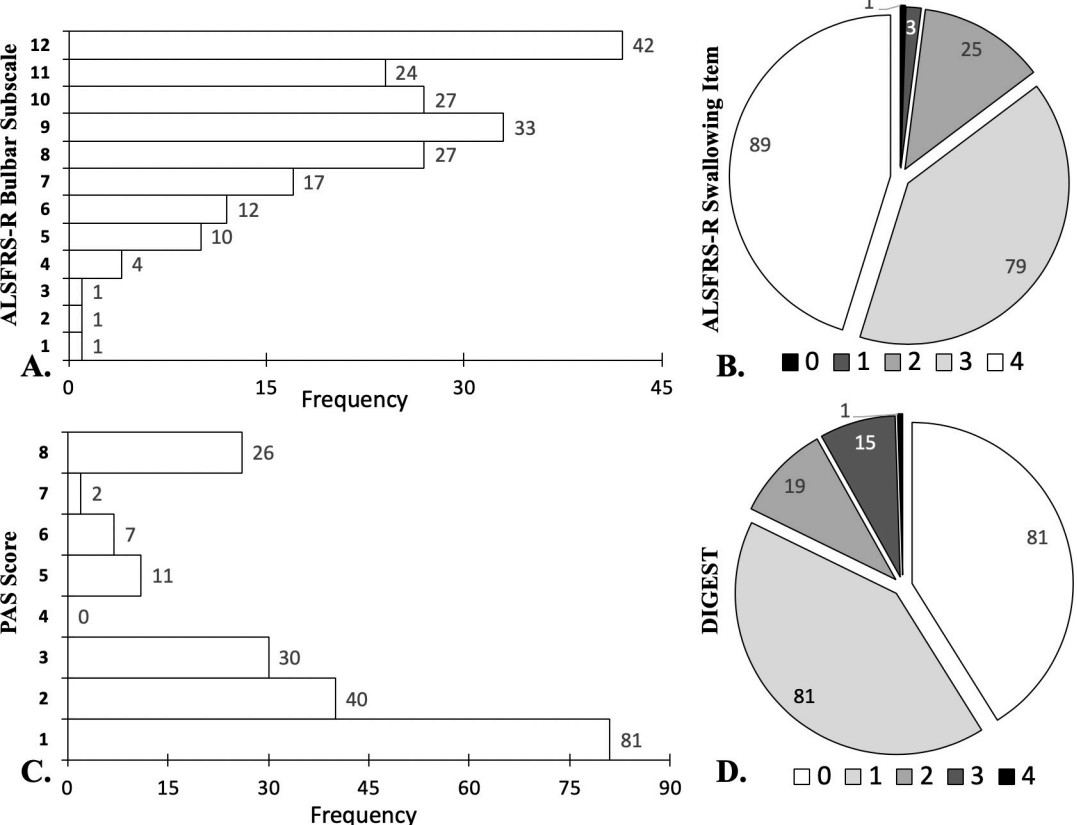

**Fig 3. Frequency distribution of outcomes of interest in 197 participants.** A) Amyotrophic Lateral Sclerosis Functional Rating Scale–Revised (ALSFRS-R) Bulbar Subscale scores. B) ALSFRS-R swallow item scores. C) Penetration aspiration scale scores (PAS). D) Dynamic Imaging Grade of Swallowing Toxicity scores (DIGEST).

development of sensitive tools to adequately screen relative risk of swallowing impairment for use in multidisciplinary ALS clinics and research settings alike.

## Bulbar subscale

Classification accuracy of the ALSFRS-R bulbar subscale to detect global pharyngeal dysphagia was considered poor to fair when comparing our results to accepted screening tool criterion levels [19]. No clear score or threshold emerged that yielded an acceptable balance between sensitivity and specificity when examining ROC outcomes across bulbar subscale scores. An effective screening tool should accurately and quickly identify at risk individuals to triage for further comprehensive evaluation and ideally minimize false negatives (i.e., missing individuals with impairment) while at the same time avoiding over identification of individuals without the disorder being screened (i.e. false positives). While generally specificity is sacrificed at the cost of increased sensitivity; a screening tool with high sensitivity but very low specificity will create undue strain on health care workers, lead to overutilization of resources and unnecessary testing, and increase patient and caregiver burden. To this end, an ALSFRS-R bulbar subscale score of ≤11 correctly identified 87% of ALS patients with global pharyngeal dysphagia; however, misclassified two-thirds of patients as dysphagic who demonstrated normal swallowing on VFSE. Use of a lower cut-point of ≤10 decreased sensitivity to an unacceptable level without significant improvements in specificity, PPV or NPV. This cut point would miss one-

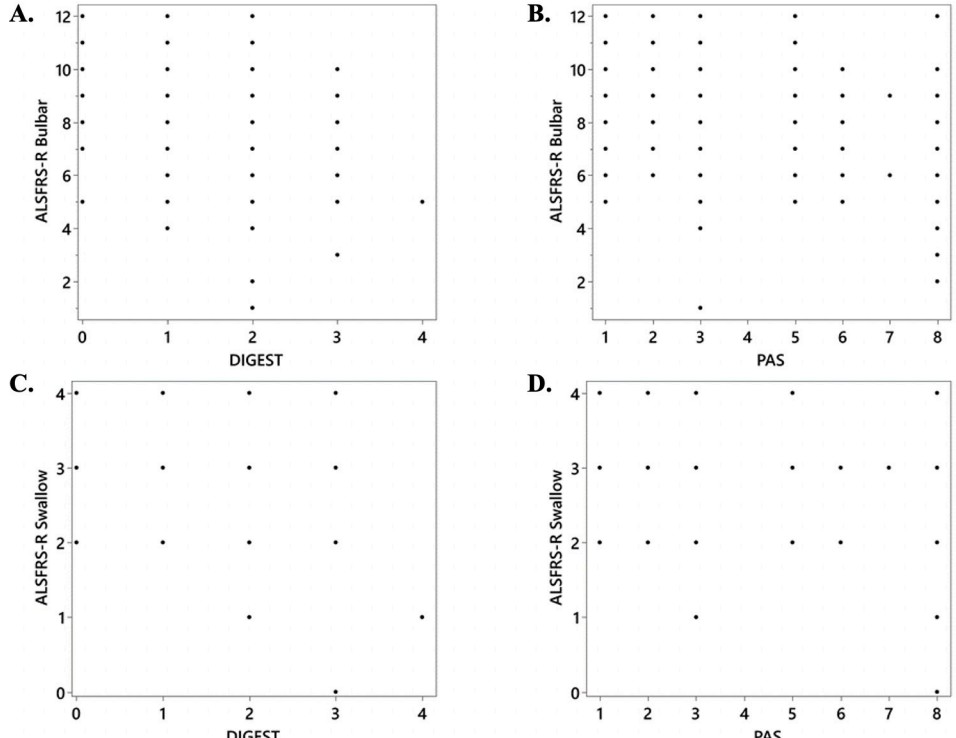

**Fig 4. Scatterplots for the amyotrophic lateral sclerosis Functional Rating Scale–Revised (ALSFRS-R) and radiographic swallowing outcomes of interest (dynamic imaging grade of swallowing toxicity–DIGEST and penetration-aspiration scale–PAS).**

quarter of patients with global dysphagia (i.e. false negatives) and would over-refer 52% of patients without dysphagia for additional testing. Finally, a bulbar subscale score of < 9 derived the most balanced degree of sensitivity and specificity of 68%, however would misclassify one-third of individuals with global pharyngeal dysphagia (false negatives).

Similarly, no cut score emerged for the ALSFRS-R bulbar subscale to detect penetration or aspiration. Although sensitivity of the bulbar subscale to detect unsafe swallowing was

**Table 1. Summary of receiver operator characteristic curve results for the ALSFRS-R bulbar subscale and swallowing item to detect unsafe swallowing.**

| Unsafe Swallowing | Bulbar Subscale: | | | | | Swallowing Item: | | |
|---|---|---|---|---|---|---|---|---|
| | ≤11 | ≤10 | ≤9 | ≤8 | ≤7 | ≤3 | ≤2 | ≤1 |
| **Sensitivity** | 92.1% | 85.5% | 77.6% | 59.2% | 43.4% | 78.9% | 26.3% | 5.3% |
| (95% CI) | (86.0, 98.2) | (77.6, 93.4) | (68.3, 87.0) | (48.2, 70.3) | (32.3, 54.6) | (69.8, 88.1) | (16.4, 36.2) | (0.2, 10.3) |
| **Specificity** | 29.8% | 45.5% | 62.0% | 77.7% | 90.1% | 60.3% | 92.6% | 100% |
| (95% CI) | (21.6, 37.9) | (36.6, 54.3) | (53.3, 70.6) | (70.3, 85.1) | (84.8, 95.4) | (51.6, 69.0) | (87.9, 97.2) | (100, 100) |
| **PPV** | 45.2% | 49.6% | 56.2% | 62.5% | 73.3% | 55.6% | 69.0% | 100% |
| (95% CI) | (37.3, 53.0) | (41.1, 58.2) | (46.7, 65.7) | (51.3, 73.7) | (60.4, 86.3) | (46.2, 64.9) | (52.1, 85.8) | (100, 100) |
| **NPV** | 85.7% | 83.3% | 81.5% | 75.2% | 71.7% | 82.0% | 66.7% | 62.7% |
| (95% CI) | (75.1, 96.3) | (74.3, 92.3) | (73.6, 89.5) | (67.6, 82.8) | (64.6, 78.9) | (74.0, 90.0) | (59.5, 73.8) | (55.9, 69.5) |
| **AUC:** | 0.76 | | | | | 0.72 | | |

Unsafe swallowing was defined as Penetration Aspiration Scale (PAS) scores ≥ 3. (CI, confidence interval, PPV, positive predictive value. NPV, negative predictive value, AUC, area under curve.)

# Penetration & Aspiration

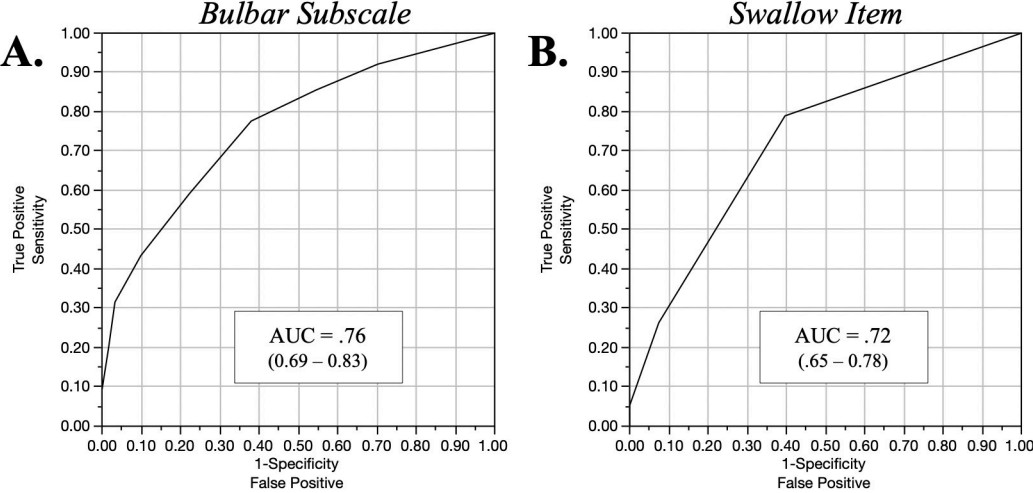

# Global Pharyngeal Dysphagia

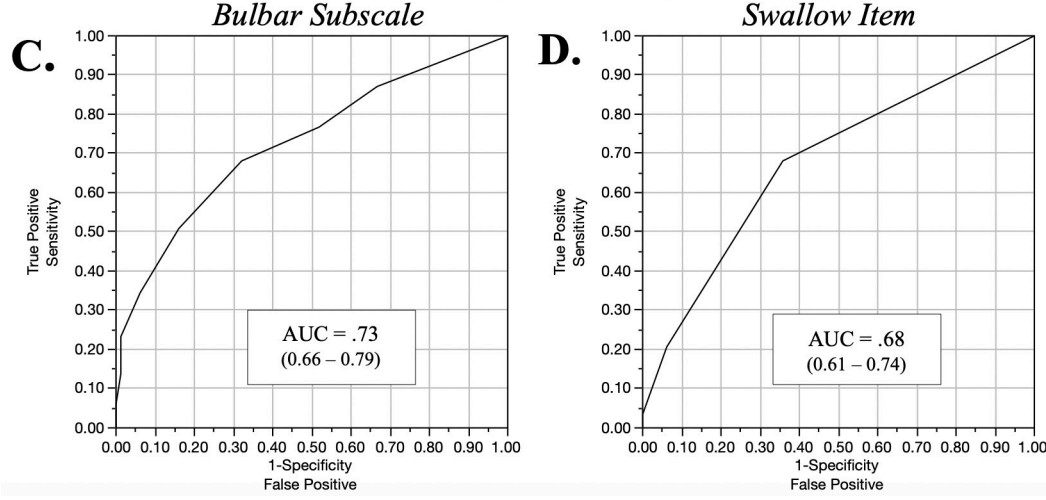

**Fig 5. Receiver operator characteristic curve results for the amyotrophic lateral sclerosis Functional Rating Scale–Revised (ALSFRS-R).** (A, B) ALSFRS-R bulbar subscale and swallowing item to detect unsafe swallowing. (C, D) Global pharyngeal dysphagia.

good-excellent for higher scores of 10 and 12 ($>$ 86%) they were associated with low specificity (30% and 45%), high false positives, and ow PPVs. These findings are in agreement with observations the bulbar subscale is not sensitive to detect *early speech* impairment in ALS patients when compared to objective physiologic speech metrics [20].

## Swallowing item

The ALSFRS-R swallowing item demonstrated poor overall screening accuracy to classify both global swallowing and safety status. Unlike the bulbar subscale, however, a clear score emerged to optimize obtained sensitivity and specificity. A swallowing item score of $\leq$ 3 accurately classified only 68% of individuals with confirmed global dysphagia missing approximately one-third of impaired individuals and representing a PPV that is not acceptable. Further, this score

**Table 2. Summary of receiver operator characteristic results for the ALSFRS-R bulbar subscale and swallowing item to detect global pharyngeal dysphagia.**

| Global Dysphagia | Bulbar Subscale: | | | | | Swallowing Item: | | |
|---|---|---|---|---|---|---|---|---|
| | ≤11 | ≤10 | ≤9 | ≤8 | ≤7 | ≤3 | ≤2 | ≤1 |
| Sensitivity | 87.1% | 76.7% | 68.1% | 50.9% | 34.5% | 68.1% | 20.7% | 3.4% |
| (95% CI) | (81.0, 93.2) | (69.0, 84.4) | (59.6, 76.6) | (41.8, 60.0) | (25.8, 43.1) | (59.6, 76.6) | (13.3, 28.1) | (0.1, 6.8) |
| Specificity | 33.3% | 48.1% | 67.9% | 84.0% | 93.8% | 64.2% | 93.8% | 100% |
| (95% CI) | (23.1, 43.6) | (37.3, 59.0) | (57.7, 78.1) | (76.0, 91.9) | (88.6, 99.1) | (53.8, 74.6) | (88.6, 99.1) | (100, 100) |
| PPV | 65.2% | 67.9% | 75.2% | 81.9% | 88.9% | 73.1% | 82.2% | 100% |
| (95% CI) | (57.7, 72.7) | (59.9, 75.9) | (67.0, 83.5) | (73.1, 90.8) | (79.7, 98.1) | (64.8, 81.5) | (69.0, 96.5) | (100, 100) |
| NPV | 64.3% | 59.1% | 59.8% | 54.4% | 50.0% | 58.4% | 45.2% | 42.0% |
| (95% CI) | (49.8, 78.8) | (47.2, 71.0) | (49.8, 69.8) | (45.7, 63.1) | (42.1, 57.9) | (48.2, 68.7) | (37.7, 52.8) | (35.0, 48.9) |
| AUC: | 0.73 | | | | | 0.68 | | |

(CI, confidence interval, PPV, positive predictive value. NPV, negative predictive value, AUC, area under curve.)

misclassified 36% of patients with confirmed normal swallowing as being dysphagic. Clearly this 'optimal' score is not acceptable for distinguishing global swallowing status in ALS.

Examination of the swallowing item's classification accuracy to differentiate safe vs. unsafe ALS swallowers was noted to be higher, with a cut score of ≤3 yielding a sensitivity of 79% and a specificity of 60%. Although improved, diagnostic utility at this optimal score threshold remained suboptimal for a useful screening tool given that it would miss one in every five penetrator/aspirators and would over-refer 40% of patients without impairment for further evaluation. An important consideration when interpreting these data is the fact that individuals with ALS may not be fully aware of subtle dietary adaptations or modifications they may implement to compensate for a progressive decline in function [21–23]. This is highly relevant given that the ALSFRS-R is a patient report outcome that asks patients to select the descriptor for a function being queried.

Given that the ALSFRS-R is commonly used in research as a baseline stratification tool or outcome to measure change in function over time; researchers are advised to consider these findings for future clinical trials. Further, clinical adoption of these scores as a dysphagia screen could create unnecessary burden for patients and their caregivers and facilitate inappropriately timed referrals for instrumental swallowing evaluations.

Although no published screening tool exists for dysphagia in ALS, two reports have examined the clinical utility of another patient-reported outcome measure (PROM) and of voluntary cough testing to detect aspiration. The Eating Assessment Tool (EAT-10) is a validated 10-item swallowing specific PROM that is available in 13 languages [24]. A cut score of >8 on the EAT-10 demonstrated a sensitivity, specificity and likelihood ratio of 86%, 72% and 3.1, respectively to detect radiographically confirmed aspiration [25]. However, the discriminant ability of the EAT-10 to detect global pharyngeal dysphagia in ALS, has not yet been established. In addition to PROMs, voluntary cough function is noted to significantly differ in individuals with ALS compared to healthy age and gender matched controls, contributing to the impaired ability to effectively expel tracheal aspirate and manage secretions in this population [26]. Given that peak expiratory flow is noted to be reduced by 50% in ALS patients with unsafe swallowing [27], voluntary cough peak expiratory flow (commonly known as peak cough flow testing) has been suggested as a screen to index one's physiologic airway defense capacity [28, 29]. Future work is needed to identify additional sensitive clinical markers in order to develop and validate a pragmatic and accurate dysphagia screening tool for use in ALS clinics [8, 9, 29].

While this work represents the first attempt to examine the discriminant ability and clinical utility of the ALSFRS-R for detecting radiographically confirmed dysphagia, limitations need

to be acknowledged. First, following typical analytic methods used in dysphagia research [30, 31], the worst PAS score was utilized to determine swallowing safety status, which may have skewed outcomes towards impairment [26]. Given that we were interested in catching early impairment however, we feel that any potential bias was warranted. Further, the global pharyngeal dysphagia metric (DIGEST scale) incorporates both the frequency and severity of penetration and aspiration and therefore mitigated potential bias for this specific outcome [17]. Second, the global dysphagia outcome only examines pharyngeal phase swallowing impairments. Therefore, our exam was specific to pharyngeal phase deficits. It is possible that a patient may have rated the ALSFRS-R swallowing item to reflect or communicate perceived impairment in the oral phase that were not detected in this study with use of the DIGEST or PAS scales. Third, given practical and ethical considerations and constraints, our sample represented individuals with mild-moderate ALS severity and bulbar dysfunction with only one patient 100% dependent on non-oral nutrition. Therefore, this cohort may not represent the complete spectrum of ALS swallowing severities. Fourth, other important non-physiologic aspects related to dysphagia such as mealtime enjoyment, mealtime duration, caregiver burden and fatigue were not indexed in this study. Finally, although these data represent the largest VFSE dataset presented to date, further work in additional patients is warranted to validate these findings.

## Conclusion

Early detection of dysphagia is paramount to guide timely clinical management decisions to mitigate or delay development of known sequalae. Given the widespread use of the ALSFRS-R to index bulbar and pharyngeal swallowing function, we aimed to determine the discriminant ability of the bulbar subscale and swallowing item to detect radiographically confirmed impairments in swallowing safety and global pharyngeal swallowing function using the gold standard VFSE. Overall accuracy of the ALSFRS-R was poor to diagnostic accuracy for swallowing safety and global pharyngeal swallow function did not meet acceptable standards across any score criteria. We therefore do not recommend use of the ALSFRS-R in isolation to screen for pharyngeal swallowing function and encourage the development of a disease specific screening tool that can accurately triage high-risk individuals for instrumental swallowing evaluation.

## Supporting information

**S1 Data.**
(XLSX)

## Acknowledgments

We are grateful for the support of Dr. John Wilkins and Nancy Wilkins and the individuals with ALS who participated.

## Author Contributions

**Conceptualization:** Emily K. Plowman.

**Data curation:** Jennifer L. Chapin, Lauren Tabor Gray, Amber Anderson, Lauren DiBiase, Raele Robison.

**Formal analysis:** Jennifer L. Chapin, Lauren Tabor Gray, Terrie Vasilopoulos, Justine Dallal York, Raele Robison.

**Funding acquisition:** Emily K. Plowman.

**Investigation:** Emily K. Plowman.

**Methodology:** Emily K. Plowman.

**Project administration:** Emily K. Plowman.

**Resources:** Emily K. Plowman.

**Supervision:** James Wymer, Emily K. Plowman.

**Visualization:** Jennifer L. Chapin, Emily K. Plowman.

**Writing – original draft:** Jennifer L. Chapin, Emily K. Plowman.

**Writing – review & editing:** Jennifer L. Chapin, Lauren Tabor Gray, Amber Anderson, Lauren DiBiase, Justine Dallal York, James Wymer, Emily K. Plowman.

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
