## [Decision Letter · Decision Letter 0]

16 Jun 2020

PONE-D-20-14426

Diagnostic utility of the Amyotrophic Lateral Sclerosis Functional Rating Scale -Revised to detect early pharyngeal dysphagia in individuals with amyotrophic lateral sclerosis.

PLOS ONE

Dear Dr. Plowman,

Thank you for submitting your manuscript to PLOS ONE. After careful consideration, we feel that it has merit but does not fully meet PLOS ONE’s publication criteria as it currently stands. Therefore, we invite you to submit a revised version of the manuscript that addresses the points raised during the review process.

We look forward to receiving your revised manuscript.

Kind regards,

Michelle Ciucci, PhD

Academic Editor

PLOS ONE

Journal Requirements:

Additional Editor Comments (if provided):

Apologies that this took so long. It is difficult to find qualified reviewers for this work and a lot of people are declining to review at this time. We appreciate your patience!

Reviewers' comments:

Reviewer's Responses to Questions

**Comments to the Author**

1. Is the manuscript technically sound, and do the data support the conclusions?

Reviewer #1: Yes

Reviewer #2: Yes

2. Has the statistical analysis been performed appropriately and rigorously? 

Reviewer #1: Yes

Reviewer #2: Yes

3. Have the authors made all data underlying the findings in their manuscript fully available?

Reviewer #1: Yes

Reviewer #2: Yes

4. Is the manuscript presented in an intelligible fashion and written in standard English?

Reviewer #1: Yes

Reviewer #2: Yes

5. Review Comments to the Author

Reviewer #1: This is a well-written manuscript that evaluated the discriminant ability for the ALSFRS-R and the question regarding swallowing ability to predict abnormal penetration-aspiration scale or DIGEST scores. The authors report poor to fair classification ability for both the ALSFRS-R and swallowing question. Comments are below.

1. Recommend changing the title to remove “early”, as recruitment did not target patients in the early stage of ALS

Abstract

2. Considering videofluoroscopy to be a gold-standard is questionable, especially given that FEES may be more sensitive to small amounts of airway invasion (e.g., https://pubmed.ncbi.nlm.nih.gov/17906496)

3. Optimized classification accuracies are mentioned in the abstract but not in the methods or results section

Introduction

4. Introduction is thorough yet concise, but is lacking a hypothesis

Methods

5. What were the exclusion criteria?

6. Please clarify if participants were given a cue to swallow during the videofluoroscopy

7. Figure 1 is a nice depiction of the PAS, but it should be amended to include the same information for the DIGEST

8. Lines 161-163: were the videofluoroscopies rated in duplicate for DIGEST only or for PAS as well? As this statement is in the DIGEST sub-section, it is not clear

9. It is mentioned in the limitations that the worst PAS scores were used, but not in the methods

Results

10. The results would be improved with scatterplots of the ALSFRS-R and swallowing question vs. the PAS and DIGEST. This will help the reader identify the relationship/lack thereof between the index test and reference standard

11. It would be beneficial for the authors to explore the false negative and false positive cases. If a patient reported swallowing difficulty on the ALSFRS-R but did not have airway invasion or residue, what was causing the difficulty? There may be merit in using the ALSFRS-R in addition to PAS/DIGEST, but it is not clear what else it may capture This could be explored further as well in the Discussion

Discussion

12. This is briefly covered, but recommend exploring what might be captured by the ALSFRS-R but missed by the PAS/DIGEST

13. Also, recommend changing use of “swallowing function” to describe PAS/DIGEST, as this is only a small part of oro-pharyngeal-esophageal swallowing function

Reviewer #2: Thank you for the opportunity to review this paper looking at the diagnostic utility of the Amyotrophic Lateral Sclerosis Functional Rating Scale-Revised. Dysphagia is an important issue for people with ALS and this paper's focus on the bulbar subscale and swallow index is valid. The sample size is good and validated outcomes measures are used. The methods and data analysis are clinically relevant and accurate. Conclusions are clinically important to disseminate. This is a well written paper and the findings are important.

Introduction - well written, appropriate reference to literature, well argued rationale for study.

Methods - how were participants enrolled? Consecutively through a clinic or by invitation? Please clarify.

Were there other exclusion criteria? Those who were solely enteral tube fed? Those unable to swallow saliva???

Has the DIGEST been used in ALS previously? Please provide this information for the reader.

Results - well presented and relevant- nice use of charts and tables

Discussion - well written and relevant. Good discussion of alternative indicators of dysphagia risk available. It would be good to discuss whether DIGEST and aspiration provide ALL the relevant dysphagia burden indicators for those with ALS. Do DIGEST and aspiration represent other aspects of burden of importance? Such as carer burden, mealtime enjoyable / mealtime impact etc...

6. PLOS authors have the option to publish the peer review history of their article (what does this mean?). If published, this will include your full peer review and any attached files.

Reviewer #1: Yes: Corinne A. Jones

Reviewer #2: No

---

## [Author Response · Author response to Decision Letter 0]

23 Jun 2020

Dear Dr. Ciucci and Reviewers,

We sincerely thank you for the expert review of our manuscript and the constructive feedback. Please find below a response to each of the reviewers’ comments or questions.

Reviewer 1:

1. We have removed the word “early” from the title as suggested.

2. Abstract: Use of the word “gold standard examination” when referring to the videofluoroscopic swallowing exam is an established reference that is supported by many in our field [1]. Additionally, other researchers have utilized the VFSS as the ‘gold standard reference tool’ in other clinical validation studies [2-5]. Our laboratory utilizes both instrumental swallowing exam (ISE) techniques and this comment is interesting given we just received comments from another submission using FEES requesting that we state it is a limitation, as the VFSS is the gold standard and FEES is not. We see the utility in use of either ISE and acknowledge that different settings may have different needs. We prefer not to get into a debate regarding which ISE is better but believe the use of this term is acceptable and, in this paper, needed to validate its use as the reference standard. We respectfully therefore stand by our use of this term.

3. We have added a statement regarding optimization of classification accuracy in both the methods and results section as advised. 

4. Introduction: We have added a hypothesis as suggested following the aims.

5. Methods: The exclusion criteria were provided in the original manuscript on page 5 in the Methods section under “Participants” we have expanded this in the revised document to read “Inclusion and exclusion criteria were: 1) confirmed diagnosis of ALS (Revised El Escorial criteria) by a neuromuscular neurology specialist; 2) not pregnant, 3) no allergies to barium, and 4) still consuming some form of foods and liquids by mouth.”

6. Methods: In response to your request to clarify if participants were given a cue to swallow during the videofluoroscopy, yes they were. Please see a reference to this in the original manuscript under “Reference Standard” – “A standardized bolus presentation was administered utilizing a cued instruction to swallow and included”

7. We have added an additional Figure to illustrate the DIGEST scale as suggested (Please see Figure 2).

8. Methods: Were the videofluoroscopies rated in duplicate for DIGEST only or for PAS as well? As this statement is in the DIGEST sub-section, it is not clear. Yes, both the PAS and DIGEST were rated in duplicate. We apologize that this was not clear and see how this may have been the case given that this description of duplicate ratings was in the DIGEST section. In the revised manuscript, we have moved this description of duplicate ratings to appear immediately after the subheading “Outcome measures” and prior to both outcomes. 

9. Methods: Thank you for pointing out the omission for using worst PAS score in the methods. We have included this in the revised manuscript under ‘VFSE outcome measures’ and subsection of ‘Swallowing Safety’ of methods.

10. Results: We have added scatterplots as suggested.

11. Results: While we agree that it would be interesting to look at what might be was causing false negative or positives (i.e., your question - If a patient reported swallowing difficulty on the ALSFRS-R but did not have airway invasion or residue, what was causing the difficulty?”), patients did not elaborate on responses but rather completed the PRO in the validated fashion by merely circling a response (0 – 4) on the scale. The study design and the information obtained using the ALSFFRS-R – prohibits our ability to answer this question and this was outside the scope of this study. 

12. Discussion: This is briefly covered, but recommend exploring what might be captured by the ALSFRS-R but missed by the PAS/DIGEST. 

We have elaborated on this point in the revised manuscript to include a statement regarding the possibility that a patient rated the ALSFRS-R swallowing item to reflect or communicate perceived impairment solely in the oral phase that were not detected in this study with the DIGEST scale.

13. Discussion: Also, recommend changing use of “swallowing function” to describe PAS/DIGEST, as this is only a small part of oro-pharyngeal-esophageal swallowing function. 

This point is well taken. We have re-worded text in the discussion that read as “swallowing function” to pharyngeal swallowing (or impairment etc) given that the safety and residue metrics using the DIGEST is a focused pharyngeal outcome that do not index oral or esophageal function. 

We thank Reviewer One for their time and expertise in helping make this manuscript improved for publication.

 

Reviewer #2: 

1. Methods - how were participants enrolled? Please clarify.

All eligible participants attending the ALS University multidisciplinary clinical were informed of this study and extended an invitation to participate. Those who agreed were enrolled. This therefore represents a convenience sample. We have provided more detailed information regarding this in the revised manuscript (please see Methods – Participants subsection).

2. Were there other exclusion criteria? Those who were solely enteral tube fed? Those unable to swallow saliva??? Inclusion and exclusion criteria were: 1) confirmed diagnosis of ALS (Revised El Escorial criteria) by a neuromuscular neurology specialist; 2) not pregnant, 3) no allergies to barium, and 4) still consuming some form of foods and liquids by mouth. This is in the revised manuscript under the Methods – Participant section.

3. Has the DIGEST been used in ALS previously? Please provide this information. Yes, it has been used previously in ALS, we have added these references in the methods section (please see Methods, Global Pharyngeal Swallowing subsection).

4. It would be good to discuss whether DIGEST and aspiration provide ALL the relevant dysphagia burden indicators for those with ALS. Do DIGEST and aspiration represent other aspects of burden of importance? Such as carer burden, mealtime impact etc. This is an excellent point. We have included in the revised manuscript, Limitations section the following new points to speak to this comment:

a. “Second, the global dysphagia outcome only examines pharyngeal phase swallowing impairments. Therefore, our exam was specific to pharyngeal phase deficits. It is possible that a patient may have rated the ALSFRS-R swallowing item to reflect or communicate perceived impairment in the oral phase that were not detected in this study with use of the DIGEST or PAS scales.”

b. Fourth, other important non-physiologic aspects related to dysphagia such as mealtime enjoyment, mealtime duration, caregiver burden and fatigue were not indexed in this study.

 

References:

1. Wirth R, Dziewas R, Beck AM, et al. Oropharyngeal dysphagia in older persons - from pathophysiology to adequate intervention: a review and summary of an international expert meeting. Clin Interv Aging. 2016;11:189-208. Published 2016 Feb 23. doi:10.2147/CIA.S97481

2. Leslie P, Drinnan MJ, Finn P, Ford GA, Wilson JA. Reliability and validity of cervical auscultation: a controlled comparison using videofluoroscopy. Dysphagia. 2004;19(4):231-240.

3. Costa MM. Videofluoroscopy: the gold standard exam for studying swallowing and its dysfunction. Arq Gastroenterol. 2010;47(4):327-328. doi:10.1590/s0004-28032010000400001

4. Szczesniak MM, Maclean J, Zhang T, et al. Inter-rater reliability and validity of automated impedance manometry analysis and fluoroscopy in dysphagic patients after head and neck cancer radiotherapy. Neurogastroenterol Motil. 2015;27(8):1183-1189. doi:10.1111/nmo.12610

5. Edmiaston J, Connor LT, Steger-May K, Ford AL. A simple bedside stroke dysphagia screen, validated against videofluoroscopy, detects dysphagia and aspiration with high sensitivity. J Stroke Cerebrovasc Dis. 2014;23(4):712-716.

---

## [Editor Report · Decision Letter 1]

30 Jun 2020

PONE-D-20-14426R1

Diagnostic utility of the Amyotrophic Lateral Sclerosis Functional Rating Scale - Revised to detect pharyngeal dysphagia in individuals with amyotrophic lateral sclerosis.

PLOS ONE

Dear Dr. Plowman,

Thank you for submitting your manuscript to PLOS ONE. After careful consideration, we feel that it has merit but does not fully meet PLOS ONE’s publication criteria as it currently stands. Therefore, we invite you to submit a revised version of the manuscript that addresses the points raised during the review process.

We look forward to receiving your revised manuscript.

Kind regards,

Michelle Ciucci, PhD

Academic Editor

PLOS ONE

Additional Editor Comments (if provided):

Thank you for the attention to the reviewer comments. There are two outstanding items. The first is the problematic term of 'gold standard.' While I agree with the authors that this is not the place for a semantics debate, when a new test outperforms an established test (that is the gold standard) the newer, better test is judged to be different than the gold standard (i.e. not as good). There are some articles (see the work of Susan Butler) that address this. Please use, 'most commonly used' or some other terminology. Second, (#11 from Reviewer 1), while it is understandable that the PRO can not provide information about false positives or negatives, this should be an important limitation that is discussed with regard to this study. There are a few more very minor comments from the second reviewers that should be addressed. Please include a note to me in the cover letter that I will address these minor changes editorially. Thank you for submitting this excellent work!

---

## [Author Response · Author response to Decision Letter 1]

3 Jul 2020

Per the email below - please refer to the cover letter, we have made both changes suggested by the reviewer and we were not sent any other edits. Thank you for your consideration. 

Dear Dr. Plowman,

Thank you for submitting your manuscript to PLOS ONE. After careful consideration, we feel that it has merit but does not fully meet PLOS ONE’s publication criteria as it currently stands. Therefore, we invite you to submit a revised version of the manuscript that addresses the points raised during the review process.

We look forward to receiving your revised manuscript.

Kind regards,

Michelle Ciucci, PhD

Academic Editor

PLOS ONE

Additional Editor Comments (if provided):

Thank you for the attention to the reviewer comments. There are two outstanding items. The first is the problematic term of 'gold standard.' While I agree with the authors that this is not the place for a semantics debate, when a new test outperforms an established test (that is the gold standard) the newer, better test is judged to be different than the gold standard (i.e. not as good). There are some articles (see the work of Susan Butler) that address this. Please use, 'most commonly used' or some other terminology. Second, (#11 from Reviewer 1), while it is understandable that the PRO can not provide information about false positives or negatives, this should be an important limitation that is discussed with regard to this study. There are a few more very minor comments from the second reviewers that should be addressed. Please include a note to me in the cover letter that I will address these minor changes editorially. Thank you for submitting this excellent work!

---

## [Editor Report · Decision Letter 2]

15 Jul 2020

Diagnostic utility of the Amyotrophic Lateral Sclerosis Functional Rating Scale - Revised to detect pharyngeal dysphagia in individuals with amyotrophic lateral sclerosis.

PONE-D-20-14426R2

Dear Dr. Plowman,

We’re pleased to inform you that your manuscript has been judged scientifically suitable for publication and will be formally accepted for publication once it meets all outstanding technical requirements.

Kind regards,

Michelle Ciucci, PhD

Academic Editor

PLOS ONE
---

## [Editor Report · Acceptance letter]

3 Aug 2020

PONE-D-20-14426R2 

Diagnostic utility of the Amyotrophic Lateral Sclerosis Functional Rating Scale – Revised to detect pharyngeal dysphagia in individuals with amyotrophic lateral sclerosis. 

Dear Dr. Plowman:

I'm pleased to inform you that your manuscript has been deemed suitable for publication in PLOS ONE. Congratulations! Your manuscript is now with our production department. 

Kind regards, 

on behalf of

Dr. Michelle Ciucci 

Academic Editor

PLOS ONE